# An isolable, chelating bis[cyclic (alkyl) (amino)carbene] stabilizes a strongly bent, dicoordinate Ni(0) complex

Braulio M. Puerta Lombardi[1], Morgan R. Faas[1], Daniel West[1], Roope A. Suvinen[2], Heikki M. Tuononen ●[2] ✉ & Roland Roesler ●[1] ✉

Chelating ligands have had a tremendous impact in coordination chemistry and catalysis. Notwithstanding their success as strongly σ-donating and π-accepting ligands, to date no chelating bis[cyclic (alkyl)(amino)carbenes] have been reported. Herein, we describe a chelating, $C_2$-symmetric bis[cyclic (alkyl) (amino)carbene] ligand, which was isolated as a racemic mixture. The isolation and structural characterization of its isostructural, pseudotetrahedral complexes with iron, cobalt, nickel, and zinc dihalides featuring eight-membered metallacycles demonstrates the binding ability of the bis(carbene). Reduction of the nickel(II) dibromide with potassium graphite produces a dicoordinate nickel(0) complex that features one of the narrowest angles measured in any unsupported dicoordinate transition metal complexes.

Since the seminal report on the first cyclic (alkyl)(amino)carbenes (CAACs: Me$_2$CAAC, menthylCAAC, and CyCAACs) in 2005[1], these exceptionally σ-donating and π-accepting ligands have had a substantial impact on coordination chemistry and catalysis[2–6]. Notable examples include the isolation of homoleptic compounds of late transition metals in low oxidation states[7–9], main-group and organic radicals[10–12], elements in unusual oxidation states[13–16], and high-performing transition metal catalysts[17–19]. The CAAC ligand palette has been expanded to incorporate other N-Dipp-substituted (Dipp = 2,6-iPr$_2$C$_6$H$_3$) five-membered representatives such as Et$_2$CAAC and AdCAAC[20], as well as FunCAACs featuring imine, amine, phosphine, or olefin pendant arms[21], six-membered CAAC-6 (CR$_2$ = CEt$_2$, cyclohexylene, menthylene, adamantylene)[22], and a bicyclic BiCAAC[23] (Fig. 1). More recently, CABC[24] and the redox-switchable fcCAAC[25] were accessed via different ring-closing protocols, opening avenues for further diversification of the field. Many more CAAC complexes have been characterized without the isolation of the corresponding free ligands[2–6].

The unique properties of chelating ligands, including the increased stability of their coordination compounds imparted by the chelate effect[26–28], as well as the relevance of the $C_2$-symmetry in enantioselective catalysis[29–31] have secured them a privileged role in coordination chemistry and catalysis. $C_2$-symmetric bis(phosphines)

2,2′-bis(diphenylphosphino)−1,1′-binaphthyl (BINAP) and ethane-1,2-diyl)bis[(2-methoxyphenyl)(phenyl)phosphane] (DIPAMP) were central to the research recognized with the 2001 Nobel Prize in Chemistry[32,33]. In contrast to the rich chemistry of chelating bis(phosphines)[34,35] and bis(NHC)s (NHC = N-heterocyclic carbene)[36–38], a chelating bis(CAAC) has remained elusive almost two decades after CAACs were first reported[1], although its potential has been probed computationally[39]. In pursuit of such species we recently described a stable ditopic trans-bis(CyCAAC) (Fig. 1)[40]. The potentially chelating cis-isomer could not be isolated or stabilized in the coordination sphere of transition metals because of its highly favorable conversion to an electron-rich olefin via a low-barrier Wanzlick-type proton-catalyzed mechanism. We reasoned that in $C_2$-symmetric bis(CAAC) 3 (Fig. 2), C=C bond formation would be precluded by the repulsive interaction of the 3-methyl substituents. The synthesis and coordination chemistry of this ligand will be reported herein.

## Results

The alkylation of N-Dipp-5,5-dimethyl-2-pyrroline, 1, with 1,3-propylene diiodide in acetonitrile, in a procedure similar to the one used to generate functionalized CAAC precursors[21], resulted in nearly quantitative formation of an equimolar mixture of $C_2$- (rac) and $C_s$-symmetric (meso) diiodide salts (Fig. 2). The desired $C_2$-isomer 2 crystallized

[1]Department of Chemistry, University of Calgary, 2500 University Drive NW, Calgary, AB, Canada. [2]Department of Chemistry, NanoScience Centre, University of Jyväskylä, Jyväskylä, Finland. ✉e-mail: heikki.m.tuononen@jyu.fi; roesler@ucalgary.ca

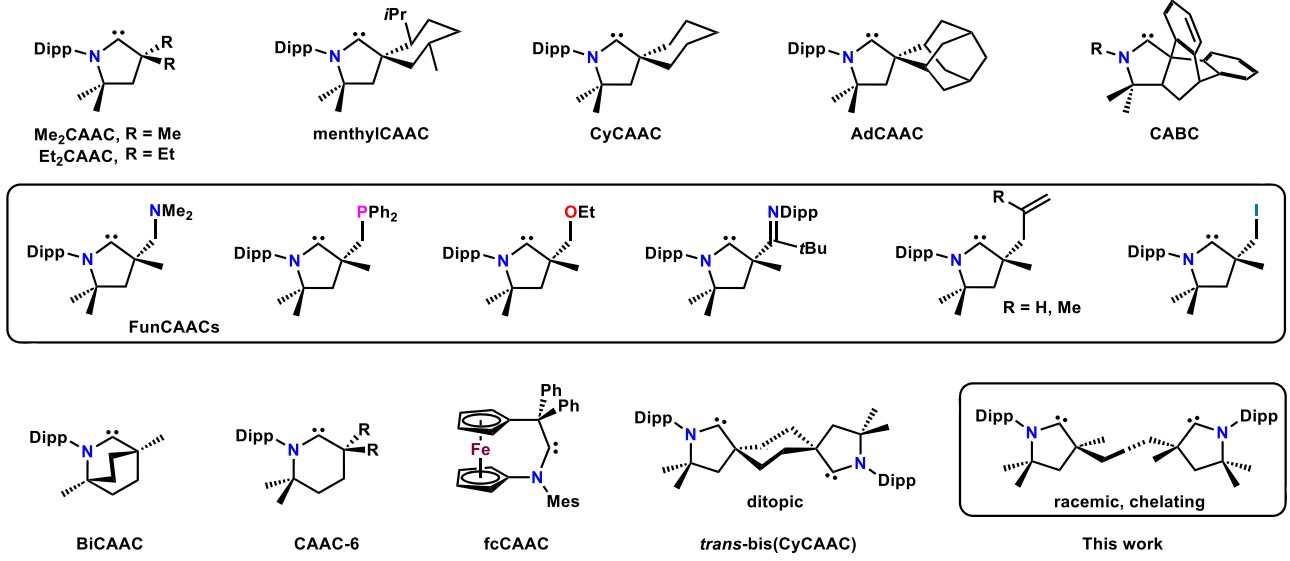

**Fig. 1 | Selected examples of isolable CAACs and their acronyms.** Dipp = 2,6-diisopropylphenyl.

**Fig. 2 | Synthesis of ligand 3 and complexes 4–8.** KHMDS = potassium hexamethyldisilazide.

preferentially from the reaction solvent as a racemic mixture and was isolated in 27% yield (*vs.* the theoretical yield of 50%).

Double deprotonation of **2** with KHMDS in THF led to the formation of the free bis(CAAC) **3** (Fig. 2), which displayed a characteristic $^{13}C$ NMR resonance corresponding to the carbene carbons at 314.7 ppm. This racemic mixture was isolated in 79% yield as a crystalline white solid that was stable under an inert atmosphere and could be stored for months at −40 °C. At room temperature, signs of decomposition became apparent in hydrocarbon solutions of **3** within 24 h. Single crystal X-ray diffraction analysis confirmed the $C_2$-symmetric bis(CAAC) structure of **3** (Fig. 3).

The coordinating ability of **3** was evaluated by reacting it with equimolar quantities of anhydrous first-row metal dihalides in THF (Fig. 2). This led to the isolation of [(**3**)MCl₂] complexes **4** (M = Fe), **5** (M = Co), and **6** (M = Zn) in 60–80% yield. A similar reaction with NiBr₂(dme) led to the formation of [(**3**)NiBr₂], **7**, which was isolated as a light blue, paramagnetic powder in 71% yield. Straightforward purification and isolation protocols exploited the low solubility of all four complexes in hydrocarbon and ethereal solvents. Crystallographic characterization revealed that the isocrystalline complexes **4–7** feature a pseudotetrahedral coordination environment at the metal ($\tau_4' = 0.96$–1.01) with a chelating ligand **3** (Fig. 4)[41]. The C−M−C angles in all four complexes fall into a narrow range between 107.68(8)−109.98(8)°. With average values of 2.051(2) and 2.404(1) Å, the Ni−C and Ni−Br bonds in **7**, respectively, are substantially longer than in square-planar (NHC)₂NiBr₂ (av. Ni−C 1.91(2) Å and Ni−Br 2.32(2)

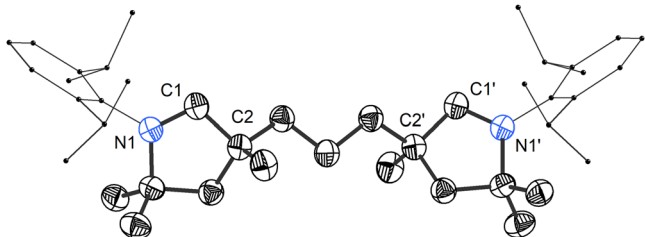

**Fig. 3 | Solid-state structure of bis(CAAC) 3 with thermal ellipsoids drawn at 50% probability and hydrogen atoms omitted for clarity.** Selected bond lengths [Å] and angles [°]: C1−N1 1.312(2); C1−C2 1.526(3); N1−C1−C2 106.01(17).

Å, 50 structures)[42] and *trans*-(Me₂CAAC)₂NiX₂ (X = Cl, Br, I; av. Ni−C 1.935(5) Å and Ni−Br 2.300(8) Å, 4 structures)[43,44]. As **3** induces a relatively weak-field splitting in the pseudotetrahedral coordination geometry it imposes[45], it allows for an interesting high-spin configuration in the case of **4**, in agreement with theoretical calculations.

Dicoordinate nickel(0) complexes are believed to be the catalytically active species in numerous transformations, most notably in C−C cross-coupling reactions[46–48]. Their activation barrier towards oxidative addition was calculated to decrease with narrowing bond angle at the metal[49], and indeed chelating phosphine ligands present advantages in nickel-catalyzed Suzuki-Miyaura cross-coupling[50–52], where they were shown to diminish off-cycle reactivity and catalyst

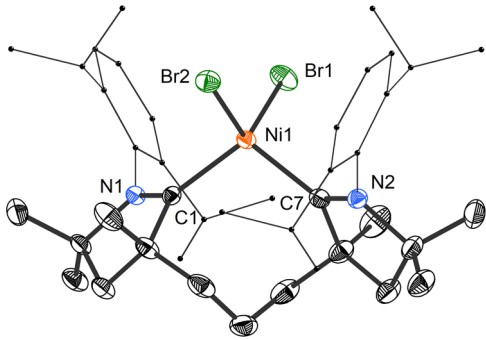

**Fig. 4 | Solid-state structure of 7 with thermal ellipsoids drawn at 50% probability and hydrogen atoms omitted for clarity.** Selected bond lengths [Å] and angles [°]: Ni1–C1 2.0491(18); Ni1–C7 2.0528(19); Ni1–Br1 2.404(3); Ni1–Br2 2.403(3); C1–N1 1.314(2); C7–N2 1.313(2); C1–Ni1–C7 107.72(7); Br1–Ni1–Br2 116.443(11).

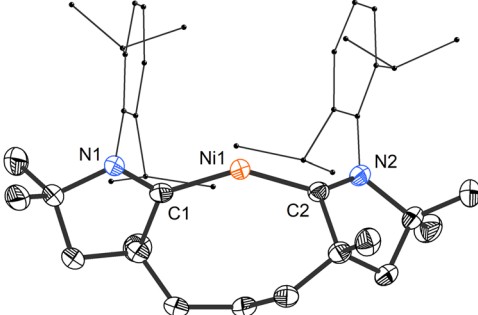

**Fig. 5 | Solid-state structure of 8 with thermal ellipsoids drawn at 50% probability and hydrogen atoms omitted for clarity.** Selected bond lengths [Å] and angles [°]: Ni1–C1 1.8209(18); Ni1–C2 1.8103(17); C1–Ni1–C2 146.70(8).

poisoning[53]. Nickel-catalyzed hydrocyanation of olefins has also benefited from the use of chelating bis(phosphines)[54–56]. While dicoordinate metal complexes are usually linear, several heavier $d^{10}$ diphosphine complexes are significantly bent, with P–M–P angles between 154.82(4) and 162.62(4)° (M = Pd, Pt)[57–59]. This is due to a combination of a flat potential energy surface for angle bending and relatively strong dispersion interactions between bulky ligands[48]. The employment of chelating bis(phosphines) allowed for the isolation of dicoordinate palladium and platinum complexes featuring even narrower P–M–P angles between 148.28(4) and 153.95(12)°[60,61]. However, no dicoordinate phosphine complexes of nickel could be isolated[62,63]. Metal-ligand orbital interactions play a key role in the bending of dicoordinate complexes of $d^{10}$ metals, and π-accepting ligands were predicted to be the best candidates for the isolation of such nickel derivatives[64]. An illustrative example in this regard is the fleeting intermediate Ni(CO)₂, with a C–Ni–C bond angle between 144.5 and 150.7°[65–67]. Consequently, the chelating ligand 3 with its two π-accepting CAACs was identified as an excellent candidate for the stabilization of a bent, dicoordinate nickel(0) complex.

Reduction of 7 with an excess of potassium graphite in THF led to a color change from pale blue to dark fuchsia. Subsequent workup yielded 8 as a black, crystalline solid exhibiting a UV-vis absorption band at 510 nm. X-ray diffraction confirmed the formation of bis(CAAC)Ni complex 8 (Fig. 5), featuring a bent geometry at the dicoordinate metal. With 146.70(8)°, the C–Ni–C angle is substantially narrower than in related homoleptic (NHC)₂Ni (>174.5(11)°)[68] and (CAAC)₂Ni complexes (166.42(5) and 164.95(15)°)[42,43], which have thus far been the most strongly bent, divalent nickel(0) derivatives. In fact, 8 features one of the narrowest angles measured in any unsupported dicoordinate transition metal complex. Most strongly bent dicoordinate transition metal complexes present additional non-bonding, stabilizing interactions involving the metal, such as π-aryl, anion–cation, and metallophilic interactions[69]. The Ni–C bonds in 8 (1.8209(18) and 1.8103(17) Å) are much shorter than those measured in the tetrahedral complex 7, and comparable to those observed in (NHC)₂Ni (av. 1.86(2) Å, 8 structures)[41] and (CAAC)₂Ni analogs (av. 1.845(3), 2 structures)[42,43]. The broad signals in the ¹H NMR spectrum of 8 suggest fluxional behavior, as reported for related complexes with chelating bis(NHC) ligands[70]. The characteristic ¹³C NMR resonances corresponding to the carbene carbons appeared at 227.2 and 234.4 ppm. A nickel(I) derivative, [(3)NiBr] (9), could also be obtained in reaction of 3 with [(IPr)NiBr]₂ and featured the expected trigonal-planar geometry at nickel (Supplementary Fig. 24).

The metal-ligand bonding in 8 was analyzed using density functional theory (DFT) and energy decomposition analysis (EDA). For comparison purposes, complexes (CAAC)₂Ni and (NHC)₂Ni with N-Dipp substituents were also considered. The results (Supplementary

Table 2) show that the metal-ligand interactions ($\Delta E_{int}$) in 8 and (CAAC)₂Ni are comparable (−633 and −644 kJ mol⁻¹, respectively), while those in (NHC)₂Ni are the weakest (−592 kJ mol⁻¹). Nevertheless, the ligand preparation energies render the bond energy ($\Delta E_{bond}$) calculated for 8 (−600 kJ mol⁻¹) in between that of (CAAC)₂Ni and (NHC)₂Ni (−634 and −565 kJ mol⁻¹, respectively). Even though the ligand 3 loses to two monodentate CAACs in terms of bond energy (and enthalpy), it triumphs in entropy through the chelate effect. Thus, the calculated Gibbs free energy change for the reaction (CAAC)₂Ni + 3 → 8 + 2 CAAC is negative, favoring 8 over (CAAC)₂Ni, though only by 6 kJ mol⁻¹.

In contrast to the $C_2$-symmetric complexes 4–7, the nickel(0) species 8 features an asymmetric ligand with noticeable differences in the orientation of the two CAAC rings with respect to the metal. Calculations probing the conformational space of 8 identified a second minimum that is $C_2$-symmetric and only 12 kJ mol⁻¹ higher in energy than the global minimum. Interestingly, the $C_2$-symmetric structure has an even narrower C–Ni–C bond angle of 133.8°, while the energetics of its metal-ligand bonding are essentially identical to that in 8. Hence, the energy difference between the two conformers stems from changes in the preparation energy of the ligand. This is in agreement with prior analyses of metal-ligand bonding in Ni(CO)₂, which showed that, while the energies of metal $d$-orbitals change significantly upon varying the C–Ni–C bond angle, the metal-ligand interaction energy $\Delta E_{int}$ draws a very shallow energy curve[63].

## Discussion

18 years after the seminal first report of a CAAC ligand, we describe the isolation of a chelating representative of this family, as a racemic mixture. The chelating ability of the $C_2$-symmetric bis(CAAC) ligand 3 was demonstrated by the characterization of its iron, cobalt, nickel, and zinc dihalide complexes 4–7. The geometric constraints imposed by 3 allowed for the stabilization of nickel(0) derivative 8, which is one of the most strongly bent, unsupported dicoordinate transition metal complexes reported to date. We expect ligand 3 to be a valuable platform for the generation of unconventional transition metal and main-group element compounds. Its applicability should be further expanded by variation of the alkyl linker scaffold and chiral resolution, and investigations to this extent are on the way.

## Methods
### General information
Unless otherwise stated, synthesis and handling of all compounds was performed under strict exclusion of air and moisture in an argon atmosphere, using a double-manifold vacuum line and an MBRAUN glove box operating with argon. Pentane was dried using an MBRAUN solvent purification system and stored in a 500 mL air-tight glass vessel containing sodium. Benzene, toluene, and tetrahydrofuran (THF) were

dried over potassium, distilled for storage into 500 mL air-tight vessels containing sodium/benzophenone ketyl, and vacuum-transferred into the reaction vessel. Acetonitrile and dichloromethane were dried over calcium hydride and stored in 500 mL air-tight vessels over 4 Å molecular sieves. 1,3-diiodopropane (Oakwood Chemicals), $FeCl_2$, $CoCl_2$, $ZnCl_2$ (Alfa-Aesar), $NiBr_2(dme)$ (Millipore-Sigma), and all other reagents (Millipore-Sigma, Oakwood Chemicals) were used as received. The enamine precursor was synthesized according to a literature procedure[21], and was passed through a silica plug in hexanes before use. Sigman's bromide dimer, $[(IPr)NiBr]_2$, was synthesized by following a reported procedure[71].

Nuclear magnetic resonance (NMR) spectra were acquired on Bruker Avance and Avance III 400 MHz spectrometers at 298 K, unless otherwise noted. $^{1}H$ and $^{13}C$ NMR chemical shifts were referenced to residual solvent peaks and naturally abundant $^{13}C$ resonances for all deuterated solvents: $CHCl_3$ (7.26 ppm, $^{1}H$) and $CHCl_3$-$d_1$ (77.16 ppm, $^{13}C$); $CH_2Cl_2$-$d_1$ (5.32 ppm, $^{1}H$) and $CH_2Cl_2$-$d_2$ (54.00 ppm, $^{13}C$); THF-$d_7$ (3.58 ppm, $^{1}H$) and THF-$d_8$ (67.21 ppm, $^{13}C$); benzene-$d_5$ (7.16 ppm, $^{1}H$) and benzene-$d_6$ (128.06 ppm, $^{13}C$)[72].

X-ray crystallographic data were collected on a Bruker SMART APEX II CCD diffractometer using suitable single crystals coated in Paratone 8277 oil (Exxon) and mounted on glass-fiber loops. Measurements were processed with the Apex III software suite. Structures were solved using the SHELXT[73] structure solution program with intrinsic phasing and refined using the SHELXL[74] refinement package with least squares minimization, all under the Olex2 platform[75]. Full crystallographic details can be found in each independently uploaded crystallographic information file (cif).

All elemental analyses were obtained on a Perkin-Elmer Model 2400 Series II analyzer. High resolution electrospray mass spectra (HRESI-MS) were obtained with a Kratos MS-80 spectrometer using samples prepared in the glovebox and transferred in a gas-tight syringe.

### Synthesis of $C_2$-symmetric bis(iminium) salt 2 (racemic)
In an argon glovebox, a 150 mL air-tight flask containing a stir bar was charged with 1,3-diiodopropane (2.00 g, 6.76 mmol), the enamine precursor 1 (5.50 g, 20.3 mmol), and anhydrous acetonitrile (15 mL). The flask was sealed and heated outside the glovebox at 100 °C for 72 h, leading to the formation of an off-white precipitate. The flask was opened to air, the mixture was filtered, and the solid was washed with acetonitrile (3 × 10 mL) followed by diethyl ether (2 × 10 mL) and dried in vacuo. Diiodide salt 2 was obtained as a white powder (1.50 g, 1.79 mmol, 27% yield). **Anal.** Calcd. for $C_{41}H_{64}N_2I_2$: C 58.71; H 7.69; N 3.34. Found: C 58.82; H 7.61; N 3.29. $^{1}H$ NMR ($CD_2Cl_2$, 25 °C, 400 MHz): $\delta$ 1.24 (d, $^{3}J_{HH}$ = 6.7 Hz, 3H, CH(C$H_3$)$_2$), 1.28 (d, $^{3}J_{HH}$ = 6.7 Hz, 3H, CH(C$H_3$)$_2$), 1.35 (d, $^{3}J_{HH}$ = 6.7 Hz, 3H, CH(C$H_3$)$_2$), 1.40 (d, $^{3}J_{HH}$ = 6.7 Hz, 3H, CH(C$H_3$)$_2$), 1.53 (s, 3H, C(C$H_3$)$_2$), 1.63 (s, 3H, C(C$H_3$)$_2$), 1.76 (s, 3H, C$H_3$), 2.07 (m, 1H, CC$H_2$C$H_2$), 2.23 (m, 2H, CC$H_2$C$H_2$), 2.41 (d, $^{2}J_{HH}$ = 13.9 Hz, 1H, C$H_2$), 2.62 (sept, $^{3}J_{HH}$ = 6.8 Hz, 1H, C$H$(CH$_3$)$_2$), 2.75 (sept, $^{3}J_{HH}$ = 6.8 Hz, 1H, C$H$(CH$_3$)$_2$), 2.83 (d, $^{2}J_{HH}$ = 13.9 Hz, 1H, C$H_2$), 7.38 (m, 2H, m-C$_6$H$_3$), 7.56 (vt, J = 7.8 Hz, 1H, p-C$_6$H$_3$), 10.51 (s, 1H, CH = N). $^{13}C$ NMR ($CD_2Cl_2$, 25 °C, 101 MHz): $\delta$ 21.9 (s, CC$H_2$CH$_2$), 22.2 (s, CH(C$H_3$)$_2$), 22.4 (s, CH(C$H_3$)$_2$), 24.7 (s, C$H_3$), 26.9 (s, CH(C$H_3$)$_2$), 27.1 (s, CH(C$H_3$)$_2$), 28.7 (s, C(C$H_3$)$_2$), 28.8 (s, C(C$H_3$)$_2$), 30.1 (s, C$H$(CH$_3$)$_2$), 30.2 (s, C$H$(CH$_3$)$_2$), 39.8 (s, CC$H_2$CH$_2$), 48.2 (s, C$H_2$), 52.4 (s, C(CH$_3$)CH$_2$), 83.8 (s, C(CH$_3$)$_2$), 125.6 (s, m-C$_6$H$_3$), 125.8 (s, m-C$_6$H$_3$), 129.0 (s, i-C$_6$H$_3$), 132.2 (s, p-C$_6$H$_3$), 144.6 (s, o-C$_6$H$_3$), 144.8 (s, o-C$_6$H$_3$), 191.7 (s, CH = N).

### Synthesis of $C_2$-symmetric bis(CAAC) 3 (racemic)
In an argon glovebox, a 50 mL round bottomed flask containing a stir bar was charged with 2 (2.00 g, 2.39 mmol) and KHMDS (0.998 g, 5.00 mmol). The flask was attached to a swivel frit and 30 mL of THF was vacuum transferred to the mixture, which was then stirred under argon for three hours. The solvent was subsequently removed in vacuo

and the residue was extracted with benzene (2 × 20 mL) and filtered. After removing benzene in vacuo, the crude product was washed with cold pentane and dried under high vacuum to yield 3 (1.11 g, 1.89 mmol, 79.1%) as a white powder. X-ray quality crystals were obtained via slow evaporation of a pentane solution. **Anal.** Calcd. for $C_{41}H_{62}N_2$: C 84.47; H 10.72; N 4.81. Found: C 84.71; H 10.88; N 4.71. $^{1}H$ NMR ($C_6D_6$, 25 °C, 400 MHz): $\delta$ 1.14 (s, 3H, C(C$H_3$)$_2$), 1.15 (s, 3H, C(C$H_3$)$_2$), 1.23 (d, $^{3}J_{HH}$ = 6.8 Hz, 3H, CH(C$H_3$)$_2$), 1.25 (d, $^{3}J_{HH}$ = 6.8 Hz, 3H, CH(C$H_3$)$_2$), 1.25 (d, $^{3}J_{HH}$ = 6.8 Hz, 3H, CH(C$H_3$)$_2$), 1.29 (d, $^{3}J_{HH}$ = 6.8 Hz, 3H, CH(C$H_3$)$_2$), 1.43 (s, 3H, C$H_3$), 1.51 (d, $^{2}J_{HH}$ = 12.9 Hz, 1H, C$H_2$), 1.79 (d, $^{2}J_{HH}$ = 12.9 Hz, 1H, C$H_2$), 1.85 (m, 1H, CC$H_2$C$H_2$), 2.00 (m, 2H, CC$H_2$C$H_2$), 3.19 (sept, $^{3}J_{HH}$ = 6.8 Hz, 1H, C$H$(CH$_3$)$_2$), 3.23 (sept, $^{3}J_{HH}$ = 6.8 Hz, 1H, C$H$(CH$_3$)$_2$), 7.15 (m, 2H, m-C$_6$H$_3$), 7.22 (vt, J = 7.6 Hz, 1H, p-C$_6$H$_3$). $^{13}C$ NMR ($C_6D_6$, 25 °C, 101 MHz): $\delta$ 21.8 (s, CC$H_2$CH$_2$), 21.9, 22.0, 25.7, 26.3, 26.4, 29.2 (s, C$H$(CH$_3$)$_2$), 29.3 (s, C$H$(CH$_3$)$_2$), 29.4 (s, C(C$H_3$)$_2$), 29.7 (s, C(C$H_3$)$_2$), 42.9 (s, CC$H_2$CH$_2$), 48.0 (s, C$H_2$), 62.5 (s, C(CH$_3$)CH$_2$), 82.2 (s, C(CH$_3$)$_2$), 123.8 (s, m-C$_6$H$_3$), 124.0 (s, m-C$_6$H$_3$), 128.1 (s, p-C$_6$H$_3$), 138.1 (s, i-C$_6$H$_3$), 146.1 (s, o-C$_6$H$_3$), 146.3 (s, o-C$_6$H$_3$), 314.7 (s, C$_{carbene}$).

### Synthesis of iron(II) complex 4
In an argon glovebox, a 25 mL round bottom flask containing a stir bar was charged with 3 (0.2 g, 0.341 mmol) and anhydrous $FeCl_2$ (0.043 g, 0.339 mmol). The flask was attached to a swivel frit apparatus and 10 mL of THF was vacuum transferred on top of the solids. The mixture was stirred under argon overnight and subsequently filtered. The solid was dried under high vacuum to yield 4 (0.151 g, 0.213 mmol, 63%) as a yellow powder. X-ray quality crystals were obtained by allowing the THF reaction filtrate to stand at room temperature overnight. Magnetic susceptibility $\mu_{eff}$ was determined to be 4.78 (4 unpaired electrons) at 298 K using Evans method. The chemical shift of $Si(SiMe_3)_4$ and a 0.013 M solution of 4 in $CD_2Cl_2$ were used. **Anal.** Calcd. for $C_{41}H_{62}N_2Cl_2Fe$: C 69.39; H 8.81; N 3.95. Found: C 69.31; H 9.19; N 3.86. **HRMS** (ESI) m/z: [M + H]$^{+}$ Calcd. for $C_{41}H_{62}N_2Cl_2Fe$ 709.3712; Found: 709.3715.

### Synthesis of cobalt(II) complex 5
In an argon glovebox, a 25 mL round bottomed flask containing a stir bar was charged with dicarbene 3 (0.2 g, 0.341 mmol) and anhydrous $CoCl_2$ (0.044 g, 0.339 mmol). The flask was attached to a swivel frit apparatus and 10 mL of THF was vacuum transferred on top of the solids. The mixture was stirred under argon overnight and subsequently filtered. The solid was dried under high vacuum to yield 5 (0.147 g, 0.206 mmol, 61%) as a blue powder. X-ray quality crystals were obtained by allowing the THF reaction filtrate to stand at room temperature overnight. Magnetic susceptibility $\mu_{eff}$ was determined to be 4.27 (3 unpaired electrons) at 298 K using Evans method. The chemical shift of $Si(SiMe_3)_4$ and a 0.02 M solution of 5 in $CD_2Cl_2$ were used. **Anal.** Calcd. for $C_{41}H_{62}N_2Cl_2Co$: C 69.09; H 8.77; N 3.93. Found: C 68.89; H 9.01; N 3.82. **HRMS** (ESI) m/z: [M + H]$^{+}$ Calcd. for $C_{41}H_{62}N_2Cl_2Co$ 712.3695; Found: 712.3700.

### Synthesis of zinc(II) complex 6
In an argon glovebox, a 25 mL round bottom flask containing a stir bar was charged with dicarbene 3 (0.205 g, 0.349 mmol) and anhydrous $ZnCl_2$ (0.048 g, 0.352 mmol). The flask was attached to a swivel frit apparatus and 10 mL of THF was vacuum transferred on top of the solids. The mixture was stirred under argon overnight and subsequently filtered. The solid was dried under high vacuum to yield 6 (0.200 g, 0.278 mmol, 80%) as a white powder. X-ray quality crystals were obtained by allowing the THF reaction filtrate to stand at room temperature overnight. **Anal.** Calcd. for $C_{41}H_{62}N_2Cl_2Zn$: C 68.47; H 8.69; N 3.89. Found: C 68.35; H 9.13; N 3.79. $^{1}H$ NMR ($CD_2Cl_2$, 25 °C, 400 MHz): $\delta$ 1.20 (d, $^{3}J_{HH}$ = 6.6 Hz, 3H, CH(C$H_3$)$_2$), 1.30 (d, $^{3}J_{HH}$ = 6.4 Hz, 3H, CH(C$H_3$)$_2$), 1.32 (s, 3H, C(C$H_3$)$_2$), 1.38 (d, $^{3}J_{HH}$ = 6.4 Hz, 3H, CH(C$H_3$)$_2$), 1.38 (s, 3H, C(C$H_3$)$_2$), 1.51 (d, $^{3}J_{HH}$ = 6.6 Hz, 3H, CH(C$H_3$)$_2$),

1.61 (m, 1H, CCH$_2$CH$_2$), 1.65 (s, 3H, CH$_3$), 1.71 (d, $^2$J$_{HH}$ = 12.9 Hz, 1H, CH$_2$), 1.81 (m, 1H, CCH$_2$CH$_2$), 2.09 (m, 1H, CCH$_2$CH$_2$), 2.16 (d, $^2$J$_{HH}$ = 12.9 Hz, 1H, CH$_2$), 2.74 (sept, $^3$J$_{HH}$ = 6.4 Hz, 1H, CH(CH$_3$)$_2$), 2.82 (sept, $^3$J$_{HH}$ = 6.6 Hz, 1H, CH(CH$_3$)$_2$), 7.24 (m, 2H, m-C$_6$H$_3$), 7.39 (vt, J = 7.73 Hz, 1H, p-C$_6$H$_3$). $^{13}$C NMR (CD$_2$Cl$_2$, 25 °C, 101 MHz): δ 24.6 (s, CH(CH$_3$)$_2$), 24.7 (s, CH(CH$_3$)$_2$), 26.0 (s, CCH$_2$CH$_2$), 28.0 (s, CH(CH$_3$)$_2$), 28.3 (s, CH(CH$_3$)$_2$), 28.8 (s, CH(CH$_3$)$_2$), 29.4 (s, C(CH$_3$)$_2$), 29.8 (s, CH$_3$), 30.1 (s, CH(CH$_3$)$_2$), 30.9 (s, C(CH$_3$)$_2$), 38.4 (s, CCH$_2$CH$_2$), 43.6 (s, CH$_2$), 60.7 (s, C(CH$_3$)CH$_2$), 83.1 (s, C(CH$_3$)$_2$), 124.5 (s, m-C$_6$H$_3$), 125.4 (s, m-C$_6$H$_3$), 129.1 (s, p-C$_6$H$_3$), 134.1 (s, i-C$_6$H$_3$), 145.5 (s, o-C$_6$H$_3$), 146.9 (s, o-C$_6$H$_3$), 255.4 (s, C$_{carbene}$).

## Synthesis of nickel(II) complex 7

In an argon glovebox, a 25 mL round bottomed flask containing a stir bar was charged with dicarbene **3** (0.5 g, 0.852 mmol) and NiBr$_2$(dme) (0.263 g, 0.852 mmol). The flask was attached to a swivel frit apparatus and 15 mL of THF was vacuum transferred on top of the solids. The mixture was stirred under argon overnight and subsequently filtered. The solid was dried under high vacuum to yield **7** (0.483 g, 0.603 mmol, 71%) as a light blue powder. X-ray quality crystals were obtained via slow-diffusion of pentane into a solution of **7** in 1,2-difluorobenzene. Magnetic susceptibility μ$_{eff}$ was determined to be 3.04 (2 unpaired electrons) at 298 K using Evans method. The chemical shift of Si(SiMe$_3$)$_4$ and a 0.012 M solution of **7** in CD$_2$Cl$_2$ were used. **Anal.** Calcd. for C$_{41}$H$_{62}$N$_2$Br$_2$Ni: C 61.44; H 7.80; N 3.50. Found: C 61.44; H 8.07; N 3.41. **HRMS** (ESI) m/z: [M + H]$^+$ Calcd. for C$_{41}$H$_{62}$N$_2$Br$_2$Ni 801.2686; Found: 801.2659.

## Synthesis of nickel(0) complex 8

In an argon glovebox, a 25 mL round bottom flask containing a stir bar was charged with **7** (0.355 g, 0.443 mmol) and KC$_8$ (0.180 g, 1.33 mmol). The mixture was attached to a swivel frit and transferred to the vacuum line where it was cooled to −78 °C. THF (15 mL) was subsequently vacuum transferred on top of the solids and the mixture was allowed to warm up to room temperature over 30 minutes. The dark magenta solution was then stirred at room temperature for an additional 30 min, after which the volatiles were removed in vacuo. The residue was extracted with pentane (5 × 10 mL), the solution was filtered, and the solvent was removed in vacuo to yield **8** (0.186 g, 0.290 mmol, 65%) as a microcrystalline black powder. X-ray quality crystals of **8** were obtained via slow evaporation of a pentane solution. **Anal.** Calcd. for C$_{41}$H$_{62}$N$_2$Ni: C 76.75; H 9.74; N 4.37. Found: C 76.59; H 9.69; N 4.34. $^1$H NMR (THF-$d_8$, 25 °C, 400 MHz): δ 0.95 (d, $^3$J$_{HH}$ = 6.7 Hz, 3H, CH(CH$_3$)$_2$) 1.09 (d, $^3$J$_{HH}$ = 6.8 Hz, 3H, CH(CH$_3$)$_2$), 1.10 (s, 3H, CH$_3$), 1.14 (s, 3H, C(CH$_3$)$_2$), 1.23 (s, 3H, C(CH$_3$)$_2$), 1.26 (d, $^3$J$_{HH}$ = 6.8 Hz, 3H, CH(CH$_3$)$_2$), 1.72 (d, $^3$J$_{HH}$ = 6.8 Hz, 3H, CH(CH$_3$)$_2$), 1.87 (s, 2H CH$_2$), 2.91 (sept, $^3$J$_{HH}$ = 6.8 Hz, 1H, CH(CH$_3$)$_2$), 3.16 (sept, $^3$J$_{HH}$ = 6.7 Hz, 1H, CH(CH$_3$)$_2$), 6.90 (dd, $^3$J$_{HH}$ = 7.5 Hz, $^3$J$_{HH}$ = 1.4 Hz, 1H, m-C$_6$H$_3$), 7.00 (dd, $^3$J$_{HH}$ = 7.5 Hz, $^3$J$_{HH}$ = 1.4 Hz, 1H, m-C$_6$H$_3$)7.39 (vt, J = 7.5 Hz, 1H, p-C$_6$H$_3$). The highly fluxional signals of the propyl linker were resolved at low temperature. $^1$H NMR (THF-$d_8$, −88 °C, 400 MHz): δ 1.44 (m, 1H, CCH$_2$CH$_2$), 1.62 (m, 2H, CCH$_2$CH$_2$), 2.07 (m, 1H, CCH$_2$CH$_2$), 2.94 (m, 1H, CCH$_2$CH$_2$), 3.33 (m, 1H, CCH$_2$CH$_2$). $^{13}$C NMR (THF-$d_8$, −88 °C, 101 MHz): δ 20.7 (s, CCH$_2$CH$_2$), 22.7, 22.8, 22.9, 23.7, 25.9, 26.0, 26.7, 27.6, 28.3, 28.7 (s, CH(CH$_3$)$_2$), 28.8, 28.9, 29.3 (s, CH(CH$_3$)$_2$), 30.1 (s, CH(CH$_3$)$_2$), 30.4 (s, CH(CH$_3$)$_2$), 30.5, 45.7 (s, CCH$_2$CH$_2$), 47.2 (s, CH$_2$), 54.8 (s, CH$_2$), 60.2 (s, C(CH$_3$)CH$_2$), 60.6 (s, C(CH$_3$)CH$_2$), 68.2, 74.6 (s, C(CH$_3$)$_2$), 75.7 (s, C(CH$_3$)$_2$), 124.0 (s, m-C$_6$H$_3$), 124.3 (s, m-C$_6$H$_3$), 124.5 (s, m-C$_6$H$_3$), 124.5 (s, m-C$_6$H$_3$), 127.6 (s, p-C$_6$H$_3$), 128.1 (s, p-C$_6$H$_3$), 138.6 (s, i-C$_6$H$_3$), 139.0 (s, i-C$_6$H$_3$), 145.1 (s, o-C$_6$H$_3$), 145.6 (s, o-C$_6$H$_3$), 145.7 (s, o-C$_6$H$_3$), 146.1 (s, o-C$_6$H$_3$), 227.2 (s, C$_{carbene}$), 234.4 (s, C$_{carbene}$).

## Synthesis of nickel(I) complex 9

In an argon glovebox, a 25 mL round bottom flask containing a stir bar was charged with **3** (0.163 g, 0.278 mmol) and Sigman's bromide dimer (0.146 g, 0.139 mmol). Toluene (8 mL) was vacuum transferred on top of the solids and the solution was stirred overnight. The brown mixture was subsequently transferred to a swivel frit and filtered on the vacuum line. The solid was washed with pentane (5 mL) and dried under vacuum to yield **9** (0.137 g, 0.190 mmol, 69%) as a maroon, microcrystalline solid. X-ray quality crystals were obtained via slow diffusion of pentane into a THF solution of **9**. Magnetic susceptibility μ$_{eff}$ was determined to be 2.20 (1 unpaired electron) at 298 K using Evans method. The chemical shift of Si(SiMe$_3$)$_4$ and a 0.014 M solution of **9** in CD$_2$Cl$_2$ were used. **Anal.** Calcd. for C$_{41}$H$_{62}$N$_2$BrNi: C 68.25; H 8.66; N 3.88. Found: C 68.33; H 9.01; N 3.91. **HRMS** (ESI) m/z: [M + H]$^+$ Calcd. for C$_{41}$H$_{62}$N$_2$BrNi 640.4289; Found: 640.4261.

## Data availability

The data generated and analyzed during this study are included in this Article and its Supplementary Information. All data are also available from the corresponding authors upon request. Metrical data for the solid-state structures of **3**−**9** in this paper have been deposited at the Cambridge Crystallographic Data Centre under reference numbers CCDC 2277362–2277368, respectively. Copies of the data can be obtained free of charge from www.ccdc.cam.ac.uk/structures/. All other data supporting the findings of this study are available within the article and its Supplementary Information. This includes the coordinates of the optimized structures as source data in form of an xyz file. Source data are provided in this paper.

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

## Acknowledgements

Financial support was provided by the Universities of Calgary and Jyväskylä, as well as the Natural Sciences and Engineering Research Council of Canada (NSERC) in the form of Discovery Grant #2019-07195 to R.R. The project received funding from the European Research Council under the EU's Horizon 2020 programme (grant #772510 to H.M.T.).

## Author contributions

B.M.P.L. designed the synthesis and carried out the experimental work and data analysis, with assistance from M.R.F. and D.W., under the guidance of R.R., who managed the project; B.M.P.L. and R.A.S. carried out the computational studies, under the guidance of H.M.T.; the interpretation of the results and writing of the manuscript was carried out jointly by B.M.P.L., H.M.T., and R.R.

## Competing interests

The authors declare no competing interests.
