## [Peer Review File · Nature Communications]

An isolable, chelating bis[cyclic (alkyl)(amino)carbene] stabilizes a strongly bent, dicoordinate Ni(0) complexEditorial Note: This manuscript has been previously reviewed at another journal that is not operating a transparent peer review scheme. This document only contains reviewer comments and rebuttal letters for versions considered at Nature Communications.

Reviewers' Comments:

Reviewer #1:

Remarks to the Author:

I support publishing this work with NatCommun. My congratulations again to this fine work.

That said, I am surprised by (some) of the authors' replies. As such, I encourage the authors to reflect on and potentially reconsider their judgement in these cases.

1.) Chart 1 still focuses on various essentially equivalent CAACs (e.g., aliphatic MeCAAC, EtCAAC, CyCAAC, AdCAAC with essentially equivalent steric profile and equivalent electronic profile; the menthyl-CAAC indeed shows peculiar reactivity in some cases) and (among others) some quite exotic ones, whereas the relevant substitutions of functionalized CAACs are not indicated and abbreviated as "R". In a contribution reporting a bisCAAC, this is misleading - particularly for the non-expert reader of a general journal (note in this regard that also CH₂-bridged CAAC-NHC hybrids do not afford the free biscarbene ACIE 2016,55,12886, which btw. further highlights the value and beauty of the present contribution).

2.) The authors argue that the arrow notation is meaningful, the "best" and "easiest" depiction, which serves a diverse leadership well, as it is "commonly" used in the literature. I disagree. Just because arguably flawed things can be found in (some parts of) the literature, it does not make things scientifically better. In detail and e.g., the authors refer later to a manuscript by Stalke, Frenking, and Roesky (arguably all rooted in heavier main group chemistry, where pi-backbonding is usually weak, and all not really the youngest), which includes discussion of backbonding (<https://doi.org/10.1002/chem.201500758>; Table 2). Backbonding (out-of-plane + in-plane negative-hyperconjugative) amounts to 36.3% (21.6% + 14.7) of the overall E(orb). This is more than in many conventional Fischer-carbene complexes (note, CAACs are "aza-Fischer" Carbenes), and even many TM carbonyls. This indicates rather an issue of scholarship than what the authors call "bonding intricacies" and "widely accepted" (which is also not true, just consider the lack of manuscripts with an arrow depiction of low-valent TM-CO complexes, also referred to in the letter). In the SI (Table S2), the authors omit the contribution of pi-backbonding in the EDA, although Fig. S28 indicates a significant contribution. As such, the arrow depiction for the carbene complex in 8 is just not appropriate, and I am certain it will also not age well in the literature and/or bode well for the standing of the authors in some parts of the community. Generally, including the various MO-channels to Table S2 would add to the manuscript as it would gauge the differences between CAACs and NHCs (likely, the major part is the out-of-plane pi-backbonding contribution, cf. Fig. S28)

3.) Optional: Instead of adapting a Walsh diagram from the literature, I would find it more appropriate to construct the MO diagram from the DFT data already at hand

4.) General comment on letter: Instead of elaborating on trivialities such as the (lack of) comparability of absolute energies obtained from various codes, it would suit the authors better to have a look at the "Guides for Authors" of other pertinent journals such as ACS Inorganic Chemistry - and where the computational part in the current SI submitted to NatCommun does still not reach the quality threshold (which I personally consider acceptable though):

https://publish.acs.org/publish/author_guidelines?coden=inocaj#supporting_information

Reply to reviewers

Reviewer #1 (Remarks to the Author):

I support publishing this work with NatCommun. My congratulations again to this fine work.

That said, I am surprised by (some) of the authors' replies. As such, I encourage the authors to reflect on and potentially reconsider their judgement in these cases.

1.) Chart 1 still focuses on various essentially equivalent CAACs (e.g., aliphatic MeCAAC, EtCAAC, CyCAAC, AdCAAC with essentially equivalent steric profile and equivalent electronic profile; the menthyl-CAAC indeed shows peculiar reactivity in some cases) and (among others) some quite exotic

ones, whereas the relevant substitutions of functionalized CAACs are not indicated and abbreviated as “R”. In a contribution reporting a bisCAAC, this is misleading - particularly for the non-expert reader of a general journal (note in this regard that also CH₂-bridged CAAC-NHC hybrids do not afford the free biscarbene ACIE 2016,55,12886, which btw. further highlights the value and beauty of the present contribution).

*** We thank the reviewer for the positive assessment of the experimental work. As requested, all FunCAACs, which had been represented by only one general ChemDraw diagram, are now explicitly drawn-out in Fig. 1 (formerly Chart 1).

2.) The authors argue that the arrow notation is meaningful, the “best” and “easiest” depiction, which serves a diverse leadership well, as it is “commonly” used in the literature. I disagree. Just because arguably flawed things can be found in (some parts of) the literature, it does not make things scientifically better. In detail and e.g., the authors refer later to a manuscript by Stalke, Frenking, and Roesky (arguably all rooted in heavier main group chemistry, where pi-backbonding is usually weak, and all not really the youngest), which includes discussion of backbonding (<https://doi.org/10.1002/chem.201500758>; Table 2). Backbonding (out-of-plane + in-plane negative-hyperconjugative) amounts to 36.3% (21.6% + 14.7) of the overall E(orb). This is more than in many conventional Fischer-carbene complexes (note, CAACs are “aza-Fischer” Carbenes), and even many TM carbonyls. This indicates rather an issue of scholarship than what the authors call “bonding intricacies” and “widely accepted” (which is also not true, just consider the lack of manuscripts with an arrow depiction of low-valent TM-CO complexes, also referred to in the letter).

*** There are at least 6 different ways for the representation of a CAAC-Metal bond in the scientific literature of the last 18 years, neither of which reflects all bonding intricacies. We favor what we believe to be the simplest and oldest way to represent a donor-acceptor bond, the arrow representing the donation of an electron pair. (Strong) back-bonding certainly exists, but simple bond drawing models generally do not reflect the existing back-bonding, which is implicitly understood (see metal carbonyls). Also, we do not find that the representation of the CAAC ligand with a “bow” serves well to reflect strong back-bonding. To conclude, we are partial to a simple bonding model in general and in complex 8 in particular, have discussed the subject matter with recognized experts in the field, and hope all members of the scientific community will understand the reasoning behind this decision, even if they prefer a different representation.

In the SI (Table S2), the authors omit the contribution of pi-backbonding in the EDA, although Fig. S28 indicates a significant contribution. As such, the arrow depiction for the carbene complex in 8 is just not appropriate, and I am certain it will also not age well in the literature and/or bode well for the standing of the authors in some parts of the community. Generally, including the various MO-channels to Table S2

would add to the manuscript as it would gauge the differences between CAACs and NHCs (likely, the major part is the out-of-plane pi-backbonding contribution, cf. Fig. S28)

*** The pi-contribution was not omitted but is included in the orbital interaction term. Rather, it cannot be separated from the sigma-contribution because the systems in question were described using atomic fragments (owing to the nature of the bis-CAAC ligands) that themselves were calculated as spherical spin-restricted singlets without symmetry constraints to ensure that the same reference state was obtained for all studied complexes (to allow comparison).

3.) Optional: Instead of adapting a Walsh diagram from the literature, I would find it more appropriate to construct the MO diagram from the DFT data already at hand

*** We argue in favour of the simple schematic Walsh diagram and consider it much easier to follow than similar diagrams based on the actual MOs of the complexes. We also think that the information in the Walsh diagram is easily used to interpret the trends in HOMO - HOMO-4 orbitals shown in Supplementary Fig. 28 and, thereby, follow the discussion in the SI.

4.) General comment on letter: Instead of elaborating on trivialities such as the (lack of) comparability of absolute energies obtained from various codes, it would suit the authors better to have a look at the "Guides for Authors" of other pertinent journals such as ACS Inorganic Chemistry - and where the computational part in the current SI submitted to NatCommun does still not reach the quality threshold (which I personally consider acceptable though): https://publish.acs.org/publish/author_guidelines?coden=inocaj#supporting_information

*** As instructed in the ACS guidelines, computational data relevant to the results of the study were already provided in the SI. We have now added some technical details to the text and highlight that default settings were used for all programs unless otherwise stated. The choice of the functional-basis sets combination is also better justified to meet the ACS guidelines.